# Fish Oil Microcapsules as Omega-3 Enrichment Strategy: Changes in Volatile Compounds of Meat Products during Storage and Cooking

**DOI:** 10.3390/foods10040745

**Published:** 2021-04-01

**Authors:** Juan Carlos Solomando, Teresa Antequera, Alberto Martín, Trinidad Perez-Palacios

**Affiliations:** 1Research Institute of Meat and Meat Products (IProCar), University of Extremadura, Avda. de las Ciencias s/n, 10003 Cáceres, Spain; juancarlosg@unex.es (J.C.S.); tantero@unex.es (T.A.); 2Nutrition and Food Science, Institute of Agri-Food Resources (INURA), University of Extremadura, Avda. Adolfo Suárez s/n, 06007 Badajoz, Spain; amartinunexes@gmail.com

**Keywords:** volatile compound, ω-3 PUFA microcapsule, cooked sausage, dry-cured sausage, storage, cooking

## Abstract

This work aims to analyze the effects of processing and storage on the volatile compound profile of different meat products enriched in ω-3 polyunsaturated fatty acids (PUFA). Monolayered (Mo) and multilayered (Mu) microcapsules of fish oil were tested. The profiles of volatile compounds were analyzed by solid-phase microextraction (SPME) coupled with gas chromatography-mass spectrometry (GC-MS). The enrichment with Mo significantly increases the abundance of volatile compounds from lipid oxidation and markers of ω-3 PUFA oxidation, which may be related to the multilayer structure of chitosan–maltodextrin in Mu that achieves greater fish oil protection than the simple coating of maltodextrin in Mo. Besides, the changes in volatile compounds during storage depends on the type of fish oil microcapsules and the meat products, having an increased abundance of ω-3 PUFA oxidation markers in dry-cured sausages added with Mo. However, the enrichment of these meat products with Mo and Mu does not modify the usual variations in the volatile compound profile during culinary cooking. Thus, the addition of multilayer fish oil microcapsules may be a suitable option for enrichment of meat products in ω-3 PUFA without modifying the abundance of volatile compounds, including oxidation markers.

## 1. Introduction

Meat is recognized by the content of nutrients of high biological value, mainly proteins [1]; however, the lipid profile of meat and meat products is often not well-regarded because of the content of saturated fatty acids (SFA), which may be high in some products, and the ratio of ω-6/ω-3 polyunsaturated fatty acids (PUFA) [2]. These components are related with an increase in the risk of suffering certain metabolic and cardiovascular diseases and different types of cancer [3,4,5,6]. Besides, a diet rich in ω-3 PUFA, mainly eicosapentaenoic acid (EPA; C20: 5 ω-3) and docosahexaenoic acid (DHA; C22:6 ω-3), is associated with the prevention and therapy of various chronic disorders, particularly those related to cardiovascular diseases [7]. In this sense, one of the current challenges for the meat industry is the improvement of the lipid profile of meat products [8], adapting to current health recommendations (250 mg EPA+DHA/person/day) established by different professional organizations and health agencies [9,10,11,12]. Moreover, there is a labelling regulation regarding nutritional claims on ω-3 PUFA: “source of ω-3 fatty acids” (minimum 48 mg EPA+DHA/100 g and 100 kcal) and “high in ω-3 fatty acids” (minimum 80 mg EPA+DHA/100 g and 100 kcal) [13].

Consequently, some previous works evaluated the possibility of incorporating oils rich in ω-3 PUFA, as an emulsion or directly, in some foods [14,15,16]. Nevertheless, due to the existence of double bonds, ω-3 PUFA are highly susceptible to oxidation under pro-oxidant conditions, i.e., at high temperatures or in contact with oxygen, light or iron [17,18]. Lipid oxidation leads to primary oxidation products, mainly hydroperoxides, which are isomerized and degraded to volatile compounds. Oxidation of ω-3 PUFA may lead to detrimental sensory attributes [19,20,21]. Besides, the development of volatile compounds providing pleasant characteristics but being potentially harmful, such as furan and its derivatives, may also take part in this enriching of food with ω-3 PUFA.

Consequently, the microencapsulation of oils rich in ω-3 PUFA has been used as a possible strategy to avoid the oxidation of these beneficial compounds [22,23,24]. The basis of this technique is to build an impediment [17,18] to avoid contact and reactivity between the active compounds and oxidative promoters of the environment, and to minimize the perception of undesirable odors and flavors [25]. In addition, microencapsulation is easy to apply and economical [26].

In meat products, the generation of volatile compounds takes place through a complex process that involves the interactions of numerous factors, proteins, lipids and additives (salt, curing agents, aromas and spices) being the main precursors and influenced by the manufacturing process [27]. Storage time and conditions also influence the development of volatile compounds related to unpleasant flavors and loss of food quality, with rancidity being identified as the main cause of flavor deterioration and low acceptability by consumers [28]. In fact, the loss of sensory quality in meat products can be monitored through the profile of volatile compounds [29].

The influence of the addition of ω-3 PUFA microcapsules to meat products has been preferably evaluated by means of the percentage of fat and fatty acids, lipid oxidation and acceptability analyses [30,31,32,33], paying less attention to the effect on the profile of volatile compounds, which may be related to the sensory attributes and/or have harmful effects, or its changes during the processing or storing.

Therefore, identifying the effect of processing and storage condition on the generation of volatile compounds in different meat derivatives enriched in omega-3 with fish oil microcapsules was the aim in the present work, evaluating two types of fish oil microcapsules.

## 2. Material and Methods

### 2.1. Experimental Design

Figure 1 outlines the experimental design of the present study. Three batches of cooked (C-SAU) and dry-cured sausages (D-SAU) were elaborated: a control (C-SAU-Co and D-SAU-Co), and with the additions of Mo (C-SAU-Mo and D-SAU-Mo) and Mu microcapsules (C-SAU-Mu and D-SAU-Mu). Mo and Mu were added in excess to label C-SAU and D-SAU as “source of ω-3 fatty acids” [13]: 3 (2.75%, *w*/*w*) and 5 (5.26%, *w*/*w*) g microcapsules/100 g product, respectively. This was because the content of EPA+DHA was different in Mo (2.75 mg/100 g) and Mu (5.26 mg/100 g) [34].

The effects of storage and culinary heating were evaluated. Thus, C-SAU batches were sampled immediately after elaboration (T0), after being refrigerated at 2–4 °C for four months (T4) and before (Be) and after (Af) being heated (90 °C, 3 min). Therefore, there were 12 C-SAU batches (Co-T0-Be; Co-T0-Af; Co-T4-Be; Co-T4-Af; Mo-T0-Be; Mo-T0-Af; Mo-T4-Be; Mo-T4-Af; Mu-T0-Be; Mu-T0-Af; Mu-T4-Be; and Mu-T4-Af). D-SAU batches were sampled immediately after elaboration (T0) and after being stored at ambient temperatures (18–20 °C), resulting in six D-SAU batches (Co-T0; Co-T4; Mo-T0; Mo-T4; Mu-T0; and Mu-T4). All batches were evaluated by means of volatile compounds. Analyses were carried out in triplicate.

### 2.2. Fish Oil Microcapsules

Elaboration of Mo and Mu microcapsules was carried out according to [34]. Emulsions (with fish oil, lecithin, maltodextrin and water (for Mo) or 1% of chitosan in acetic acid 1% (for Mu)) were prepared, high-pressure homogenized and dried by spray-drying.

### 2.3. Meat Products

Elaboration of C-SAU and D-SAU batches was carried out in a meat industry (remaining anonymous), as previously described in [35]. Therefore, the exact amounts of ingredients were not reported.

### 2.4. Analysis of Volatile Compounds

Solid-phase microextraction (SPME) and gas chromatography (GC) with a mass selective detector were applied to carry out the analysis of volatile compounds, according to [36], by using a cross-linked carboxen/polydimethylsiloxane fiber (10 mm long, 100 µm thick, Supelco, Bellefonte, PA, USA) and a Hewlett-Packard 6890 series II GC coupled to a mass selective detector (HP 5973) (Hewlett-Packard, Wilmington, DE, USA) with a 5% phenyl-95% polydimethylsiloxane column (30 m × 0.32 mm ID, 1.05 μm film thickness, Hewlett-Packard). Operating conditions were: 6 psi of column head pressure, 1.3 mL min^−1^ of flow at 40 °C, and a splitless mode. The analyses were carried out at random for two weeks. Compounds were identified by comparison of the mass spectrum with a database (National Institute of Standards and Technology (NIST) and Wiley libraries) and with linear retention (LRI) indexes available in the literature [19,21,37,38,39,40,41,42,43]. For calculating the LRI of the samples of the present work, n-alkanes (Sigma R-8769, Barcelona, Spain) were run under the same conditions. Results are expressed in area units (AU).

### 2.5. Statistical Design

Replicate experimental samples (*n* = 3) of three batches of each meat product (C-SAU and D-SAU) were analyzed in triplicate. One-way analyses of variance (ANOVA) and the Tukey’s test (if *p* < 0.05) were applied to evaluate the effects of (i) enrichment, (ii) storage and iii) culinary heating. Data on volatile compounds showing significant differences were analyzed by principal component analysis (PCA). For that, the original data were normalized and the variables were classified in the first two components. The IBM SPSS Statistics v.22 program was used.

## 3. Results and Discussion

### 3.1. Profile of Volatile Compounds in D-SAU and C-SAU Batches

The results of D-SAU and C-SAU samples show a total of 53 and 76 volatile compounds, respectively, which have been classified into different chemical families (Figure 2 and Figure 3, respectively), and Table 1 and Table 2 show the abundance of individual volatile compounds, respectively.

In D-SAU batches, acids was the most abundant chemical family of volatile compounds (Figure 2), with aldehydes, esters and terpenes being in second place, and aliphatic hydrocarbons, aromatics, alcohols, ketones, furans and cyclic hydrocarbons showed percentages lower than 5%. As for individual volatile compounds (Table 1), acetic acid had the highest abundance, followed in decreasing order by hexanal, methyl hexanoate, β-myrcene, pentanoic acid and butanoic acid, with the rest of the volatile compounds having an abundance lower than 10 AU × 10^6^. This finding agrees with results found by other authors in similar products [29,43,44,45,46]. The microbial fermentation of carbohydrates may explain the high content of acids (acetic, pentanoic and butanoic acids) [45,47]. Another characteristic compound of carbohydrate fermentation, 3-hydroxy-2-butanone, was also detected in D-SAU batches [48]. Moreover, the existence of β-myrcene, α-phellandrene and D-limonene was also noticeable and could be ascribed to the addition of spices [49].

In C-SAU (Figure 3), the most abundant family of volatile compounds was aldehydes, followed by cyclic and aliphatic hydrocarbons. Lower percentages were found for alcohols, esters, acids, ketones, terpenes, aromatics, furans and pyrazines, in decreasing order. This is similar to the profile previously found in other studies of cooked sausages [40,50]. α-thujene was the most abundant volatile compound in C-SAU, followed by pentanal, β-thujene and hexanal (Table 2). Other authors have found the highest abundance in hexanal, followed by heptanal, pentanal, benzaldehyde, nonanal alcohols (1-octen-3-ol, 1-pentanol, 2-ethyl-1-hexanol and 1-heptanol) and terpenes (limonene and β-myrcene and γ-terpinene) [50,51,52], as observed in the present work. Nevertheless, the abundance of α and β-thujene in this study was higher in comparison to previous studies. Since α and β-thujene are characteristic of medicinal herbs, essential oils, flavorings and spices, such as nutmeg [53,54], their high abundance in the samples of the present study may be ascribed to the addition of spices to C-SAU batches.

The profile of volatile compounds in D-SAU and C-SAU remained largely the same in control and enriched batches. Nevertheless, some statistical differences were detected in the percentages of chemical families (Figure 2 and Figure 3, respectively) and in the abundance of individual volatile compounds (Table 1 and Table 2, respectively). The most marked effects are described as follows. Overall, in D-SAU, lower percentages of aldehydes, terpens, esters and aliphatic hydrocarbons, and higher percentages of acids were found in D-SAU-Mu than in the other two batches; and both enriched batches showed higher percentages of furans and lower percentages of esters and aromatics than D-SAU-Co. In C-SAU, most chemical families of volatile compounds showed higher percentages in the enriched batches than in C-SAU-Co. Despite these differences, the addition of Mo and Mu fish oil microcapsules only influenced significantly the abundance of 13 and 14 of the 53 volatile compounds identified in D-SAU at T0 and T4, respectively, and of 27 and 30 of the 76 volatile compounds identified in C-SAU at T0 and T4, respectively. In general, Mo-added meat products had the highest abundance of volatile compounds from fatty acid oxidation, such as pentanal, hexanal and 1-pentanol [39], and of characteristic markers of ω-3 PUFA oxidation, such as 2-ethylfuran, 2,4-decadienal and 2-decenal [54]. Other foods (mayonnaise and nuggets) added fish oil have also shown these volatile compounds [54,55], which are related to rancid flavor and ω-3 PUFA oxidation. This effect can be ascribed to the type of fish oil microcapsule with different wall materials, maltodextrin in Mo and chitosan–maltodextrin in Mu, with the multilayer structure of chitosan–maltodextrin giving a higher protection than the simple coating of maltodextrin of Mo.

In the study of Yang et al. [20], who carried out a chemometric analysis to characterize volatile compounds from oxidized ω-3 PUFA rich oils, three groups of volatile compounds were differentiated depending on their relationship with the degree of oxidation of these oils: not related (3-hexenal, hexanal, 1,4-octadien-1-ol and 2,6-nonadienal), an intermediate relationship (ethylbenzene, 1,2-dimethylbenzene, cyclohexanone, 1,3-dimethylbenzene, 1-ethyl-4-methyl benzene, 2-penylfuran, decane and undecane) and high-quality markers of the oxidation of ω-3 PUFA rich oils (2,4-heptadienal and 2-propenal). In fact, these authors made a point to identify 2,4-heptadienal as the most sensitive marker volatile compound of ω-3 PUFA rich oils oxidation. In the samples of this study, 2,4-heptadienal was not found, which may disprove a powerful ω-3 PUFA oxidation, even when Mo was added to the meat products. However, other specific oxidation markers associated with rancid flavors in fish oil such as 2,4-decadienal [19] were found in the present study in both the C-SAU and D-SAU enriched batches

In the work of Resconi et al. [56], with raw meat stored under high oxygen conditions, eight volatile compounds were proposed as shelf-life markers (pentanoic, hexanoic and heptanoic acids, 1-hexanol, 2-undecenal, ethyl octanoate, 2-heptanone and 2-pentyl furan). In the samples of the present study, some of these markers (decane, undecane, pentanoic acid, 1-hexanol, 2-heptanone, 2-penylfuran) were found, with enriched samples showing a significantly lower abundance of three of them (pentanoic acid and 2-heptanone in C-SAU and 2-pentylfuran in D-SAU) in comparison to the control batches. The reason behind this finding may be the use of maltodextrin as a wall material in the fish oil microcapsules, since a high antioxidant capacity of this carrier agent was proved previously in gelatin powders of golden goatfish [57]. This fact is worth noting in the case of the 2-pentylfuran, because furan and its derivatives, apart from providing pleasant characteristics, are associated with potential harmful effects [58,59]

### 3.2. Storage Effect on the Profile of Volatile Compounds of D-SAU and C-SAU Enriched with Fish Oil Microcapsules

In D-SAU, the storage for 4 months at ambient temperatures did result in significant differences in any family of volatile compounds (Figure 2). However, the analysis of individual volatile compounds showed significant differences in 23 compounds (Table 1), finding a similar trend in Co, Mo and Mu batches. The abundance of pentane, decane, 1-propanol, 1-penten-3-ol, 1-pentanol, 4-terpineol, pentanal, hexanal, heptanal, nonanal, 2-decenal and 3,5-octadien-2-one increased from T0 to T4, and 4-terpineol, 2-heptanone, 2-octanone, phellandrene, D-limonene, acetic acid, butanoic acid, methylpropil acetate, benzaldehyde, humulene and ally sulphide decreased from T0 to T4. A similar pattern of changes was previously observed during the refrigeration storage of ripened sausages [29]. Most of the volatile compounds with increased abundance from T0 to T4 were lipid oxidation products, which agrees with the increases in the lipid oxidation values from T0 to T4 previously observed in the same samples [60]. The decrease in the abundance of acetic and butanoic acids, which are generated from the microbial fermentation of carbohydrates, could be explained by the gradual reduction in the homofermentative activity of staphylococci and lactic acid bacteria [61]. In fact, this finding has been previously observed in similar studies of dry-cured sausages [29,46]. Moreover, the decreases in volatile compounds generated from spices, belonging to the terpene (α-phellandrene and d-limonene) and cyclic hydrocarbons families (humulene) agree with previous studies on ripened sausages stored at refrigeration temperatures [29,62].

Figure 4a represents a bi-plot of the PCA of the volatile compounds data from D-SAU samples as affected by enrichment and storage. The first principal component (PC1) comprised 42.11% of the total variance, and the second principal component (PC2) accounted for 20.73%. The score plot indicates a clear classification of samples as a function of the enrichment and the storage: those with high negative PC1 (D-SAU-Co-T0, in the axis; D-SAU-Mo-T0, far from the axis and D-SAU-Mu-T0, very far from the axis), those with high positive PC2 (D-SAU-Co-T4), those with high positive PC1 (D-SAU-Mu-T4) and those with low positive PC2 (D-SAU-Mo-T4). As for the loading plot, 2,4-decadienal, hexanal, 1-penten-3-ol and 2-ethylfuran are located in the in the upper right quadrant, which corresponds to low positive charges in PC2, and near to D-SAU-Mo-T4. These volatile compounds come from fish oil oxidation and have, in general, low threshold odors [63,64], which may negatively influence the odor and flavor of the samples. On the other hand, pentanal and allyl-sulphyde are located in the low left quadrant, which corresponds to high negative charges in PC1, associated with D-SAU-Co-T0, D-SAU-Mo-T0 and D-SAU-Mu-T0. This is quite in agreement with differences found in individual volatile compounds (Table 2) and is consistent with the changes during storage. The association of PUFA oxidation markers with D-SAU-Mo-T4 reinforces our previous hypothesis, pointing out the more protective effect against the reactivity of fish oil of the wall of Mu (chitosan and maltodextrin) than of Mo (maltodextrin). Despite these results on the profile of volatile compounds, no marked differences were found in the sensory quality among D-SAU batches as affected by refrigeration storage [65].

In C-SAU, the storage at refrigeration showed significant differences in six families of volatile compounds (Figure 3), showing higher percentages at T4 of aliphatic hydrocarbons, alcohols and terpenes, and lower percentages of aldehydes, ketones and esters than at T0. The analysis of individual volatile compounds showed significant differences in 23 compounds (Table 2). In Co, Mo and Mu batches, the abundance of most of them increased from T0 to T4: pentane, decane, 1-propanol, 1-hexanol, 1-octen-3-ol, propanal, 2-methylpropanal, butanal, 3-methylbutanal, pentanal, heptanal, 2-octenal, 2-decenal, 2,4-decadienal, 3-heptanone, 2-ethylfuran, terpene, butanoic acid, decanoic acid and benzaldehyde, while 2-heptanone, terpinolene and 4-methyl-phenol decreased from T0 to T4. A similar pattern of changes was previously observed in irradiated cooked pork sausages, vacuum packed and stored for 4 and 8 days at refrigeration temperature [66]; however, the comparison is complicated due to the existence of great differences between meat products, i.e., composition, processing and storage conditions. As in D-SAU, most of the volatile compounds with increased abundance from T0 to T4 were lipid oxidation products [67], which agrees with the increases in the lipid oxidation values from T0 to T4 previously observed in the same samples [60]. However, those volatile compounds previously identified in bulk fish oil [41,68] and related to PUFA oxidation and rancid flavor perceptions, such as 2,4-heptadienal, 3,5-octadien-2-one and 1-octen-3-one, have not been detected in any of the batches analyzed at T0 or T4. This is in concordance with the volatile profiles of Mo and Mu fish oil microcapsules [54]. The storage also significantly increased the abundance of acetic, butanoic and nonanoic acids, which could be explained by an increase in the growth rate of lactic acid bacteria (LAB). These bacteria are psychotrophic, microarophilic and able to resist high concentrations of salt and smoke [69]. In fact, LAB have been identified in previous studies as the main population of microorganisms in cooked vacuum-packed meat emulsions stored at refrigerated temperatures [69,70].

Figure 4b represents a bi-plot of the PCA of the volatile compounds data from the C-SAU samples as affected by enrichment and refrigeration storage. The first principal component (PC1) comprised 35.77% of the total variance, and the second principal component (PC2) accounted for 23.40%. The score plot indicates a clear classification of samples as a function of the enrichment and the storage: those with high negative PC2 (C-SAU-Mu-T0 and C-SAU-Mo-T4), those with high positive PC2 (C-SAU-Co-T0 and C-SAU-Mo-T0), those with high positive PC1 (C-SAU-Mu-T4) and those with low positive PC2 (C-SAU-Co-T4). In this meat product, according to the loading plot, volatile compounds markers of fish oil oxidation, such as propanal, pentanal, hexanal, 2-decenal and 2-ethylfuran, were not specifically associated with any batch. Thus, in contrast to D-SAU, in C-SAU the different wall material in Mo and Mu fish oil microcapsules seems not to have an effect during the refrigeration storage. In fact, a previous study found different effects on meat products of the addition of these types of microcapsules depending on the meat matrix [71].

### 3.3. Culinary Heating Effect on the Profile of Volatile Compounds of Cooked Sausages Enriched with Fish Oil Microcapsules

In C-SAU, the culinary heating significantly influenced 45 and 47 volatile compounds at T0 and T4, respectively (Table 2), with the Co, Mo and Mu batches experiencing similar changes. In general, culinary heating increased aliphatic hydrocarbons, alcohols, aldehydes, acids and furans, while esters and ketones decreased. These changes are quite in agreement with previous studies [59,72,73], and are associated with the thermal degradation of lipids, the Maillard reactions and the interaction between the products of the Maillard reaction with the lipids’ oxidized products [74].

Previous results [60] showed an increase in lipid oxidation values after the culinary cooking, with low values that varied in a narrow range (0.18–0.54 mg MDA/Kg sample). In this respect, it is worth mentioning that the numerical increase in the abundance of the most volatile compounds during culinary heating was quite scarce, even in those markers of PUFA oxidation. This may be ascribed to the short period of time required for the culinary heating of C-SAU samples, 3 min at 90 °C, since time was described as a greater influence than temperature in the development of volatile compounds during cooking [72].

## 4. Conclusions

The general profile of volatile compounds in cooked and dry-cured sausages remained largely the same in control and ω-3 PUFA enriched batches, but the type of fish oil microcapsule influenced the generation of individual volatile compounds, especially those considered as markers of ω-3 PUFA oxidation. Thus, multilayer microcapsules (Mu) with chitosan–maltodextrin as wall materials reduced the formation of volatile compounds characteristic of ω-3 fatty acid oxidation to a greater extent than microcapsules with a simple coating of maltodextrin (Mo). This suggests a better oxidative stability of the meat product enriched with multilayer microcapsules of fish oil.

The enrichment of cooked and dry-cured sausages with different fish oil microcapsules does not modify the usual changes in the profile of volatile compounds during culinary cooking in cooked sausages. However, its impact during the refrigeration storage depends on the type of fish microcapsule and on the meat product, with a significant increase in the volatile compound indicators of ω-3 PUFA oxidation in dry-cured sausages with added monolayered fish oil microcapsules.

Therefore, the addition of multilayer fish oil microcapsules may be an applicable option to increase the content of ω-3 PUFA in dry-cured and cooked meat products without modifying the abundance of volatile compounds, including oxidation markers.

## Figures and Tables

**Figure 1 foods-10-00745-f001:**
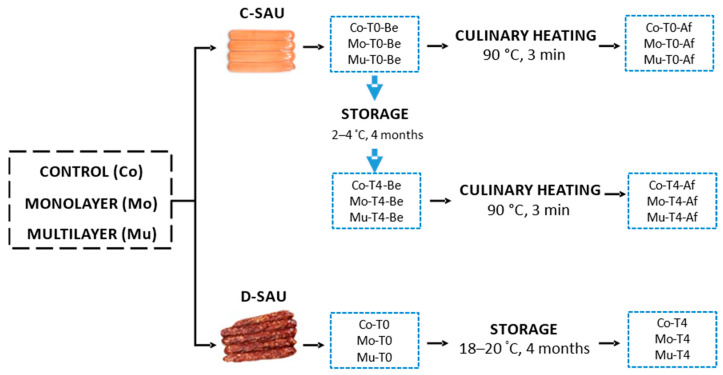
Scheme of the experimental design. C-SAU: cooked sausages; D-SAU: dry-cured sausages; Co: control batch; Mo: batch with added monolayered fish oil microcapsules; Mu: batch with added multilayered fish oil microcapsules; T0: batch sampled before storing; T4: batch sampled after storing for four months; Be: batch sampled before culinary heating; and Af: batch sampled after culinary heating.

**Figure 2 foods-10-00745-f002:**
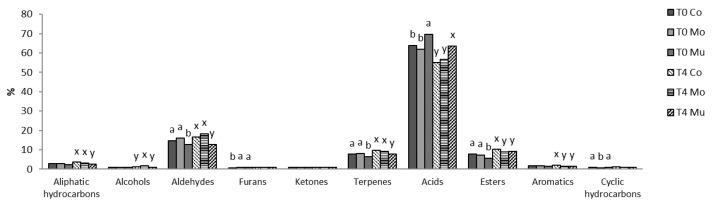
Percentage of families of volatile compounds in dry-cured sausages. Different letters indicate a significant enrichment effect among batches at T0 (a,b) and T4 (x,y). Co: control batch; Mo: batch with added monolayered fish oil microcapsules; Mu: batch with added multilayered fish oil microcapsules; T0: batch sampled before storing; T4: batch sampled after storing for four months.

**Figure 3 foods-10-00745-f003:**
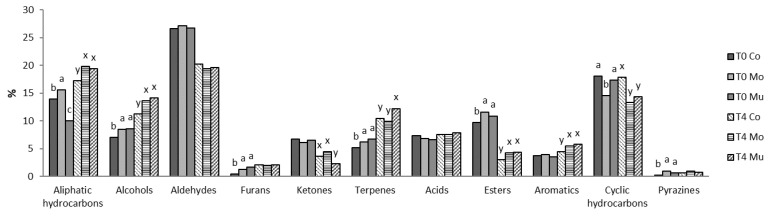
Percentage of families of volatile compounds in cooked sausages. Different letters indicate a significant enrichment effect among batches at T0 (a,b) and T4 (x,y). Co: control batch; Mo: batch with added monolayered fish oil microcapsules; Mu: batch with added multilayered fish oil microcapsules; T0: batch sampled before storing; T4: batch sampled after storing for four months.

**Figure 4 foods-10-00745-f004:**
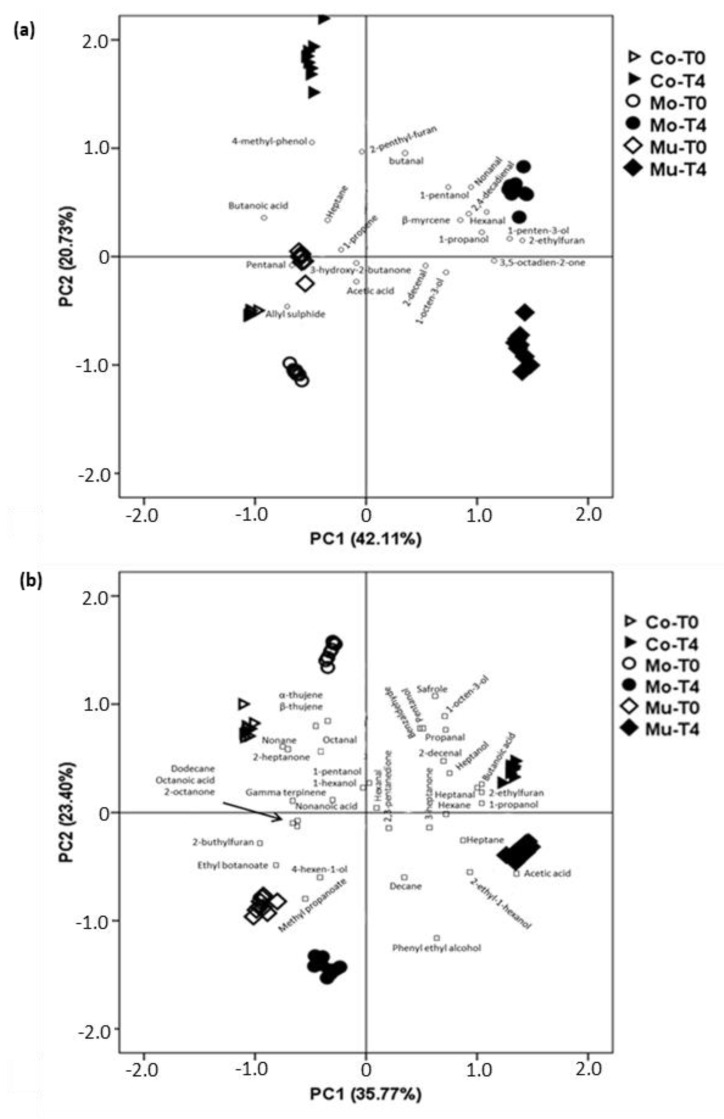
Bi-plot of principal component analysis (PCA) of volatile compounds in dry-cured (**a**) and cooked (**b**) sausages. Co: control batch; Mo: batch with added monolayered fish oil microcapsules; Mu: batch with added multilayered fish oil microcapsules; T0: batch sampled before storing; T4: batch sampled after storing for four months.

**Table 1 foods-10-00745-t001:** Storage effect (*p*S) on the abundance of volatile compounds (AU × 10^6^) in omega-3-enriched dry-cured sausages (*p*E).

LRI	ID	Chemical Group/Compound	T0			*p*E	T4			*p*E	*pS*	SEM
			Co	Mo	Mu		Co	Mo	Mu			
Aliphatic hydrocarbons												
497	A	Pentane	0.51	0.42	0.59	NS	1.51	1.47	1.31	NS	***	0.27
599	A	Hexane	0.41	0.36	0.31	NS	0.13	0.55	0.17	NS	NS	0.07
704	A	Heptane	0.42 ^b^	0.48 ^b^	0.97 ^a^	***	0.49	0.45	0.32	NS	NS	0.08
761	A	1-propene	n.d.^b^	0.54 ^a^	n.d.^b^	***	0.29 ^a^	n.d.^c^	0.13 ^b^	*	NS	0.05
800	A	Octane	6.65	6.41	6.12	NS	8.23	6.95	7.91	NS	NS	0.32
901	A	Nonane	0.54	0.42	0.61	NS	0.37	0.58	0.31	NS	NS	0.04
997	A	Decane	1.06	0.96	0.92	NS	1.29	1.39	1.34	NS	***	0.07
1097	A	Undecane	1.92	1.78	1.86	NS	1.68	1.65	1.48	NS	NS	0.10
Alcohols												
614	A	1-propanol	0.03 ^b^	0.11 ^a^	0.10 ^a^	*	0.12	0.15	0.18	NS	*	0.02
687	A	1-penten-3-ol	n.d.^b^	0.23 ^a^	n.d.^b^	***	0.31 ^c^	8.51 ^a^	6.47 ^b^	***	***	0.25
820	A	1-pentanol	1.32	1.61	1.42	NS	3.24 ^b^	4.53 ^a^	3.33 ^b^	***	***	0.06
1024	A	1-heptanol	1.08	1.13	1.10	NS	1.44	1.40	1.22	NS	NS	0.18
1031	A	1-octen-3-ol	2.78 ^c^	3.73 ^a^	3.21 ^b^	NS	2.58	3.97	3.68	NS	NS	0.20
1195	A	4-terpineol	1.85	1.88	1.79	NS	1.68	1.64	1.58	NS	**	0.03
Aldehydes												
591	A	2-methyl propanal	0.45	0.43	0.38	NS	0.30	0.37	0.29	NS	NS	0.03
618	A	Butanal	0.30 ^a^	0.09 ^b^	0.31 ^a^	**	0.36	0.52	0.20	NS	NS	0.05
667	A	2-methyl butanal	0.15	0.17	0.11	NS	0.25	0.20	0.09	NS	NS	0.03
738	A	Pentanal	7.91 ^c^	9.66 ^a^	8.73 ^b^	*	6.29 ^c^	15.40 ^a^	10.36 ^b^	***	**	0.33
862	A	Hexanal	93.84	96.14	105.06	NS	181.49 ^c^	265.60 ^a^	242.11 ^b^	***	***	6.23
942	A	Heptanal	8.01	7.19	7.25	NS	13.31	12.88	14.29	NS	***	0.62
1050	A	Octanal	0.45	0.57	0.54	NS	0.48	0.44	0.54	NS	NS	0.02
1147	A	Nonanal	5.63	5.24	5.09	NS	12.94 ^c^	14.93 ^a^	13.82 ^b^	*	***	0.13
1322	A	2-decenal	0.96	0.89	1.05	NS	0.89 ^b^	0.90 ^b^	1.34 ^a^	**	*	0.01
1395	A	2,4-decadienal	0.41	0.46	0.44	NS	0.38 ^b^	0.66 ^a^	0.42 ^b^	**	NS	0.04
Ketones												
749	A	2,3-pentanedione	0.49	0.44	0.40	NS	0.54	0.32	0.41	NS	NS	0.03
778	A	3-hydroxy-2-butanone	0.56 ^a^	0.39 ^a^	0.11 ^b^	***	0.34	0.29	0.30	NS	NS	0.04
933	A	2-Heptanone	7.00	6.99	6.58	NS	6.06	5.54	5.06	NS	**	0.27
981	A	3-Heptanone	3.51	3.88	4.69	NS	2.50	3.62	3.67	NS	NS	0.46
1039	A	2-Octanone	0.42	0.35	0.29	NS	n.d.	n.d.	n.d.	-	***	0.05
1063	A	3,5-octadien-2-one	n.d.^b^	0.11 ^a^	n.d.^b^	***	n.d.^b^	3.87 ^a^	4.21 ^a^	***	***	0.35
Furans												
722	A	2-ethyl-furan	n.d.^b^	0.15 ^a^	0.09 ^a^	*	n.d.^c^	2.84 ^a^	1.14 ^b^	***	***	0.44
837	A	3-furaldehyde	0.41	0.29	0.26	NS	0.27	0.29	0.24	NS	NS	0.02
1012	A	2-pentyl-furan	1.93 ^a^	1.19 ^b^	1.37 ^b^	**	1.60 ^a^	1.31 ^b^	1.26 ^b^	*	NS	0.06
Terpenes												
982	A	Sabinene	8.17	7.89	7.98	NS	7.71	7.49	7.56	NS	NS	0.11
1003	B	β-myrcene	36.01 ^b^	36.62 ^b^	46.78 ^a^	*	38.84	40.05	41.90	NS	NS	0.75
1021	A	α-phellandrene	5.93	6.20	6.78	NS	8.72	8.83	8.17	NS	**	0.12
1037	A	D-Limonene	3.65	3.31	3.72	NS	1.67	1.76	1.22	NS	***	0.26
1066	A	γ-terpinene	2.41	2.06	2.21	NS	2.85	2.86	2.01	NS	NS	0.10
1105	A	Terpene	1.87	1.87	1.70	NS	1.65	1.51	1.41	NS	NS	0.11
1404	A	α-cubebene	1.43	1.36	1.78	NS	1.29	1.25	1.23	NS	NS	0.09
Acids												
717	A	Acetic acid	478.69 ^b^	602.75 ^a^	452.62 ^b^	***	374.50 ^b^	439.17 ^b^	543.82 ^a^	***	*	43.89
895	A	Butanoic acid	29.05	27.74	27.48	NS	28.64 ^b^	23.33 ^a^	23.61 ^a^	**	*	1.56
986	A	Pentanoic acid	35.66	36.94	36.54	NS	36.34	36.05	35.90	NS	NS	0.40
1362	A	Nonanoic acid	2.60	2.43	2.61	NS	2.42	2.42	2.63	NS	NS	0.08
1472	A	Decanoid acid	2.87	2.78	2.61	NS	2.91	3.20	3.79	NS	NS	0.19
Esters												
786	A	Methylpropyl acetate	0.32	0.29	0.25	NS	0.19	0.19	0.13	NS	*	0.02
853	A	Methyl hexanoate	63.15	56.04	57.08	NS	63.64	63.33	73.61	NS	NS	2.76
Aromatics												
1018	A	Benzaldehyde	9.30	8.29	7.64	NS	6.05	5.81	6.14	NS	***	0.19
1190	A	4-methyl-phenol	0.53	0.59	0.61	NS	0.88 ^a^	0.57 ^b^	0.52 ^b^	**	NS	0.02
1375	A	Eugenol	4.18	4.16	4.34	NS	4.40	4.42	3.97	NS	NS	0.09
Cyclic hydrocarbons												
992	B	α-thujene	6.22	5.87	6.01	NS	6.34	6.59	6.37	NS	*	0.10
1495	A	Humulene	0.72	0.75	0.84	NS	0.35	0.26	0.27	NS	*	0.04
Other												
899	B	Allyl sulphide	2.51 ^a^	2.89 ^a^	1.94 ^b^	**	1.59	1.43	1.49	NS		0.14

^a–c^ Different letters indicate a significant enrichment effect among batches. * *p* < 0.05; ** *p* < 0.01; *** *p* < 0.001; NS: no significant; n.d., not detected; SEM, standard error; LRI, linear retention index of the compounds eluted from the GC-MS; ID, method of identification; A, mass spectrum and retention time identical with an authentic standard; and B, tentative identification by mass spectrum. Co: control batch; Mo: batch with added monolayered fish oil microcapsules; Mu: batch with added multilayered fish oil microcapsules; T0: batch sampled before storing; T4: batch sampled after storing for four months.

**Table 2 foods-10-00745-t002:** Storage (pS) and culinary heating (pC) effects on the abundance of volatile compounds (AU × 10^6^) in omega-3-enriched cooked sausages (pE).

LRI	ID	Compound	T0						*p*E	*p*C	T4						*p*E	*p*C	*p*S	SEM
			Co		Mo		Mu				Co		Mo		Mu					
			Be	Af	Be	Af	Be	Af			Be	Af	Be	Af	Be	Af				
Aliphatic hydrocarbons																				
499	A	Pentane	0.53	0.47	0.44	0.45	0.39	0.43	NS	NS	0.88	0.39	0.86	0.56	0.63	0.61	NS	*	***	0.04
601	A	Hexane	0.23	0.43	2.12	0.46	0.84	0.79	*	*	0.12	0.74	0.25	0.85	1.25	0.58	*	***	NS	0.09
695	A	1-Heptene	0.80	n.d.	1.05	n.d.	0.74	n.d.	*	***	0.85	n.d.	1.01	n.d.	1.10	n.d.	NS	***	NS	0.08
703	A	Heptane	3.35	1.55	1.54	1.13	0.75	2.08	*	NS	3.67	1.35	0.76	2.37	0.82	2.24	*	NS	NS	0.17
799	A	Octane	1.27	2.75	2.47	2.68	1.51	2.50	NS	*	1.92	2.41	2.57	2.15	2.36	1.65	NS	NS	NS	0.11
812	A	2-octene	n.d.	2.08	n.d.	2.13	n.d.	1.59	NS	***	n.d.	1.76	n.d.	1.80	n.d.	1.62	NS	***	NS	0.16
901	A	Nonane	0.23	1.91	0.46	2.15	0.32	1.23	***	***	n.d.	2.36	0.38	1.27	0.43	0.72	NS	***	NS	0.13
1000	A	Decane	n.d.	n.d.	0.19	n.d.	n.d.	0.55	*	-	2.39	0.26	2.53	0.26	2.56	0.30	NS	***	*	0.17
1101	A	Undecane	0.52	0.71	0.43	0.63	0.57	0.77	NS	NS	0.85	0.68	0.76	0.68	0.81	0.67	NS	*	NS	0.03
1200	A	Dodecane	0.42	0.43	0.27	0.49	n.d.	0.58	**	*	n.d.	0.49	0.35	0.25	0.51	0.32	**	NS	NS	0.03
1296	A	Tridecane	n.d.	n.d.	n.d.	0.31	n.d.	0.54	NS	*	n.d.	n.d.	n.d.	0.24	n.d.	0.30	-	*	NS	0.03
1402	A	Tetradecane	0.58	0.52	0.16	0.53	n.d.	0.68	NS	NS	0.85	0.50	0.19	0.68	n.d.	0.45	NS	*	NS	0.15
Alcohols																				
615	A	1-propanol	0.11	0.14	0.17	0.13	0.21	0.16	**	NS	0.18	0.28	0.69	0.40	2.09	0.42	*	NS	*	0.10
681	A	2-methyl-1-propanol	0.98	n.d.	1.24	n.d.	1.48	n.d.	NS	**	2.92	n.d.	2.68	n.d.	3.25	n.d.	NS	*	NS	0.26
825	A	1- pentanol	1.33	1.65	1.23	1.92	1.22	1.64	**	*	1.45	1.56	1.34	1.69	1.38	1.57	NS	NS	NS	0.04
923	A	1-hexanol	0.22	0.78	0.26	0.88	0.22	1.11	*	***	0.88	0.97	1.37	0.74	1.23	0.88	NS	*	*	0.06
927	A	4-hexen-1-ol	n.d.	0.98	n.d.	1.96	n.d.	1.92	*	***	n.d.	1.42	n.d.	0.98	n.d.	1.49	NS	***	NS	0.12
1024	A	1-heptanol	1.10	0.74	0.78	0.89	0.82	0.89	NS	NS	1.49	1.36	1.04	0.99	1.28	1.50	*	NS	NS	0.06
1031	A	1-octen-3-ol	0.77	3.34	0.81	3.42	0.63	2.76	*	***	2.27	5.60	3.16	5.00	2.93	6.02	*	***	***	0.31
1088	A	2-ethyl-1-hexanol	0.71	0.40	0.74	0.46	0.77	0.89	NS	*	0.60	0.56	0.82	0.89	1.06	1.30	*	NS	NS	0.14
1092	A	Phenyl ethyl alcohol	n.d.	n.d.	0.25	0.41	0.24	0.96	**	**	n.d.	n.d.	0.24	0.36	0.32	0.94	**	*	NS	0.06
Aldehydes																				
521	A	Propanal	0.16	n.d.	0.17	n.d.	0.14	n.d.	***	**	0.52	0.78	0.64	1.26	0.49	0.94	-	*	***	0.09
593	A	2-methylpropanal	1.30	1.35	1.14	1.12	1.16	1.54	NS	*	2.16	1.75	2.37	1.36	1.93	1.09	*	*	***	0.08
621	A	Butanal	0.47	n.d.	0.75	n.d.	0.38	n.d.	NS	*	1.05	n.d.	1.76	n.d.	1.59	n.d.	NS	***	*	0.11
667	A	2-methyl butanal	n.d.	0.62	n.d.	0.43	n.d.	0.54	NS	***	n.d.	0.82	n.d.	0.81	n.d.	0.76	NS	***	NS	0.07
687	A	3-methyl butanal	1.98	2.01	2.25	2.10	2.69	2.21	NS	NS	7.49	7.59	7.72	8.53	8.50	7.61	NS	NS	**	0.53
738	A	Pentanal	5.86	6.00	5.89	4.66	4.44	3.34	*	NS	5.69	6.50	6.64	6.95	6.65	7.48	*	*	*	0.61
862	A	Hexanal	4.14	4.33	4.60	3.40	4.54	3.54	NS	NS	3.29	5.57	5.12	7.56	3.86	4.92	*	*	NS	0.16
904	A	2-hexenal	0.31	n.d.	0.44	n.d.	0.26	n.d.	-	***	0.20	n.d.	0.22	n.d.	0.54	0.56	*	*	NS	0.03
939	A	Heptanal	3.18	3.50	2.26	2.77	2.69	3.13	***	*	3.98	4.33	5.49	6.26	5.24	6.84	***	***	***	0.22
1047	A	Octanal	n.d.	0.38	n.d.	0.33	n.d.	0.43	NS	***	n.d.	0.66	n.d.	0.31	n.d.	0.40	*	**	NS	0.04
1011	A	2-heptenal	0.10	n.d.	0.41	n.d.	0.11	n.d.	-	***	0.23	n.d.	0.22	n.d.	0.26	n.d.	NS	***	-	0.02
1114	A	2-octenal	n.d.	0.09	n.d.	0.12	n.d.	0.18	-	***	0.57	0.44	0.43	0.76	0.61	0.85	*	NS	***	0.05
1147	A	Nonanal	0.35	0.50	0.67	0.55	0.35	0.87	NS	NS	0.73	0.50	1.04	0.68	0.99	0.45	NS	*	NS	0.25
1223	A	2-nonenal	n.d.	n.d.	0.08	n.d.	n.d.	n.d.	-	-	n.d.	n.d.	n.d.	n.d.	0.09	n.d.	-	-	-	0.01
1286	A	2,4-nonadienal	n.d.	n.d.	0.04	n.d.	0.04	n.d.	-	-	n.d.	n.d.	0.04	n.d.	0.05	n.d.	-	-	-	n.d.
1328	A	2-decenal	0.43	0.79	0.40	1.79	0.40	1.09	*	*	0.66	1.07	1.92	3.51	0.75	1.60	*	***	*	0.09
1390	A	2,4-decadienal	n.d.	0.29	0.12	0.47	n.d.	0.37	NS	*	0.42	0.79	2.14	2.87	1.21	1.37	NS	NS	***	0.04
Ketones																				
735	A	2-pentanone	0.88	n.d.	0.78	n.d.	0.72	n.d.	-	***	0.89	n.d.	0.58	n.d.	0.64	n.d.	*	***	NS	0.07
744	A	2,3-pentanedione	n.d.	0.34	n.d.	0.47	n.d.	0.40	***	***	n.d.	0.36	n.d.	0.54	n.d.	0.43	-	***	NS	0.04
933	A	2-heptanone	1.86	0.84	1.62	0.55	1.22	0.38	*	**	1.03	1.18	0.40	0.19	0.33	n.d.	*	NS	*	0.10
979	A	3-heptanone	3.91	2.27	3.42	2.39	3.36	1.09	*	***	4.24	1.32	4.44	1.39	4.20	3.25	NS	*	*	0.23
1039	A	2-octanone	0.19	0.89	n.d.	1.35	n.d.	0.20	NS	***	0.27	0.29	0.31	0.30	0.63	0.33	***	NS	NS	0.09
1342	A	2-undecanone	0.04	n.d.	n.d.	n.d.	n.d.	n.d.	-	-	0.05	n.d.	n.d.	n.d.	0.07	n.d.	-	-	-	n.d.
Furans																				
720	A	2-ethylfuran	n.d.	0.38	n.d.	0.26	n.d.	0.21	NS	***	0.82	0.60	1.29	1.65	0.92	1.16	*	NS	***	0.08
841	A	3-furaldehyde	0.49	n.d.	0.53	n.d.	0.45	n.d.	NS	*	0.41	n.d.	0.43	n.d.	0.35	n.d.	NS	*	NS	0.04
908	A	2-butylfuran	n.d.	0.35	0.39	0.41	0.32	0.13	*	NS	0.22	0.12	0.27	n.d.	0.21	n.d.	NS	-	NS	0.03
1008	A	2-pentyl-furan	0.29	0.70	0.49	0.50	0.71	0.40	NS	NS	0.52	0.64	0.54	0.28	0.27	0.30	NS	NS	NS	0.03
Terpenes																				
1026	A	3-carene	n.d.	1.50	n.d.	1.35	n.d.	1.47	NS	***	n.d.	1.47	n.d.	1.23	n.d.	1.31	NS	***	NS	0.12
1037	A	D-limonene	0.25	2.89	0.43	2.67	0.65	2.59	NS	***	1.51	1.86	1.55	1.38	1.70	1.25	NS	NS	NS	0.13
1066	A	γ-terpinene	1.97	3.33	2.30	3.53	2.04	3.44	NS	*	2.23	3.80	2.48	2.24	1.21	2.32	*	*	NS	0.16
1097	A	Terpene	0.26	0.28	0.19	0.17	0.17	0.37	NS	NS	0.28	0.66	0.16	0.45	0.15	0.43	NS	**	*	0.08
1136	A	Terpinolene	0.70	n.d.	0.67	n.d.	0.62	n.d.	NS	**	0.39	n.d.	0.58	n.d.	0.28	n.d.	*	*	*	0.05
1116	A	β-terpinene	n.d.	n.d.	0.31	n.d.	0.40	n.d.	NS	***	n.d.	n.d.	n.d.	n.d.	0.61	n.d.	-	-	NS	0.09
1195	A	4-terpineol	n.d.	n.d.	0.32	n.d.	n.d.	n.d.	-	-	n.d.	n.d.	0.11	n.d.	n.d.	n.d.	-	-	-	0.02
1491	A	Isocayophillene	0.34	n.d.	0.26	n.d.	0.29	0.46	NS	*	n.d.	n.d.	n.d.	0.50	n.d.	n.d.	-	-	-	0.03
Acids																				
716	A	Acetic acid	n.d.	n.d.	n.d.	0.04	0.44	0.14	NS	-	n.d.	0.34	0.15	0.37	0.46	0.64	***	NS	NS	0.04
895	A	Butanoic acid	3.36	2.50	2.44	2.41	2.01	2.10	**	NS	3.43	2.14	3.47	2.90	5.19	2.98	*	**	*	0.15
898	A	2-butenoic acid	n.d.	0.27	n.d.	0.33	n.d.	0.51	-	***	n.d.	0.26	n.d.	0.17	n.d.	0.40	NS	*	NS	0.03
986	A	Pentanoic acid	1.14	1.99	0.73	1.09	0.63	0.72	***	**	0.45	1.27	0.65	1.05	0.57	0.69	NS	NS	NS	0.08
1273	A	Octanoic acid	n.d.	0.45	n.d.	0.35	n.d.	n.d.	NS	NS	n.d.	0.49	n.d.	0.42	n.d.	0.14	NS	*	NS	0.13
1366	A	Nonanoic acid	0.43	0.49	0.54	0.46	0.39	0.67	NS	NS	0.61	0.74	0.32	0.41	0.68	0.39	**	NS	NS	0.04
1461	A	Decanoid acid	0.33	0.44	0.31	0.37	0.66	0.23	*	NS	0.62	0.83	0.85	1.07	0.74	2.72	NS	*	*	0.18
Esters																				
656	A	Methyl propanoate	n.d.	0.07	n.d.	0.26	n.d.	0.37	***	***	n.d.	n.d.	n.d.	n.d.	n.d.	n.d.	-	-	-	0.02
750	A	Methyl butanoate	1.27	n.d.	1.04	n.d.	1.02	n.d.	NS	***	0.99	n.d.	1.08	n.d.	1.06	n.d.	NS	***	NS	0.20
836	A	Ethyl butanoate	2.31	2.34	2.56	2.49	2.81	2.62	**	NS	3.18	1.96	3.44	1.49	3.50	1.43	NS	***	NS	0.11
952	A	Methyl hexanoate	3.39	n.d.	3.95	n.d.	3.37	n.d.	NS	***	3.44	n.d.	3.76	n.d.	4.61	n.d.	*	***	NS	0.33
Aromatics																				
1018	A	Benzaldehyde	2.98	1.76	2.01	2.21	1.50	1.85	*	*	4.36	4.47	4.18	3.07	3.06	3.43	*	NS	***	0.19
1190	A	4-methyl-phenol	0.40	1.33	0.41	0.93	0.46	0.81	NS	*	0.24	0.23	0.38	0.36	0.36	0.50	NS	NS	*	0.18
1305	B	Safrole	0.05	0.45	0.04	0.51	0.03	0.40	-	***	0.05	0.55	0.03	0.31	0.04	0.21	*	*	NS	0.03
Cyclic hydrocarbons																				
980	B	β-thujene	5.20	4.56	3.90	2.99	4.90	3.67	*	**	4.90	4.79	4.79	2.20	5.30	2.75	*	***	NS	0.41
991	B	α-thujene	7.28	8.05	5.51	5.23	5.51	5.33	**	NS	8.26	7.18	6.13	5.17	6.45	5.91	**	*	NS	0.22
1422	B	cis-muurola-4(14),5-diene	n.d.	0.20	n.d.	0.19	n.d.	n.d.	-	-	n.d.	0.26	n.d.	n.d.	n.d.	0.24	-	.	-	0.02
1524	A	δ-cadinene	0.47	0.43	0.63	0.41	0.36	0.65	NS	NS	0.38	0.57	0.83	0.26	0.58	0.34	NS	NS	NS	0.04
Pyrazines																				
863	A	2-methylpyrazine	0.45	0.26	0.30	0.38	0.37	0.28	NS	NS	0.26	0.34	0.20	0.14	0.29	0.20	NS	NS	NS	0.11
947	A	2,6-dimethylpyrazine	n.d.	0.22	0.38	0.19	n.d.	0.24	NS	NS	n.d.	0.44	n.d.	0.41	n.d.	0.55	NS	*	NS	0.05

* *p* < 0.05; ** *p* < 0.01; *** *p* < 0.001; NS: no significant; n.d., not detected; SEM, standard error; LRI, linear retention index of the compounds eluted from the GC-MS; ID, method of identification; A, mass spectrum and retention time identical with an authentic standard; and B, tentative identification by mass spectrum. Co: control batch; Mo: batch with added monolayered fish oil microcapsules; Mu: batch with added multilayered fish oil microcapsules; T0: batch sampled before storing; T4: batch sampled after storing for four months; Be: batch sampled before culinary heating; and Af: batch sampled after culinary heating.

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
