# Peer review of "Fish Oil Microcapsules as Omega-3 Enrichment Strategy: Changes in Volatile Compounds of Meat Products during Storage and Cooking"

_foods, 2021, doi:10.3390/foods10040745_

Round 1

Reviewer 1 Report

The authors report a study that aims at improving the fatty acid profile of meat products by adding fish oil while more or less protecting the fish oil through mono- or multilayers. Given the health implications this is considered a relevant topic. Yet I recommend MAJOR REVISION for the following reasons.

To me it appears as if this manuscript more or less shows what the authors had shown earlier (https://www.mdpi.com/2304-8158/9/11/1683/htm). It should thus be made more clear why this paper should get published. Where is the novelty and relevance given also that the authors state little or no sensory differences in the products later in the discussion.

This said, to me the analysis of data and display of results should be improved: As regards the analysis itself, two-way analysis of variance including interaction effects (enrichment by storage) should be applied instead of several one-way ANOVAS. If a given sample is analysed three times, the sample ID should be included as random factor. It would especially be worth reporting the interaction effect as it indicated that the effect of storage depends on the type of enrichment.

In addition, the sample size needs clarification: How many different sausages from the same batch, how many technical replicates of the volatile analysis? As it appears to me, 3 samples were analysed three times, so n per group is 9. Thus, the statistical power of the experiment is rather low which is especially relevant when no significant differences are found/stated (which in this case could simply be due to the low n).

This said, the analysis should also focus on the magnitude of differences rather than on significance (given the low n). This could be done using e.g. volcano plots that show p vs. magnitude of differences.

Also with respect to display of data, PCA analyses should be reported/described appropriately. As is, bi-plots are shown (indicating both, scores and loadings). Description of PCA method and display of data should be revised accordingly. Plus, when interpreting the findings, it should be commented on whether or not the direction of changes during storage is consistent or not (I don’t think the bi-plots are conclusive).

This brings me to say that for the volatile analyses, it should be reported the order of analyses, i.e. how the samples were subjected to SPME analyses: all on the same day, random or systematic (i.e. batch-wise) order as this could explain the clustering of samples also. And I suggest it is clarified whether standards or only NIST database were used for identification (as it is somewhat contradictory when reading the paper).

Last but not least it is considered relevant to report the fat content and fatty acid composition of the final products and to relate this data quantitatively to the volatile data.

Finally, the proper use of English should be checked by a native speaker.

Table and Figure captions should be revised. For example, Fig 4: it says “samples analyzed before (T0) and after four months (T4) of storage at room temperature”. The latter is not true for both types of sausages! Also footer of table 1: storage of dry cured sausages was not refrigerated!

What is the point in reporting tables 1 and 2? Wouldn’t a graph showing what compounds increase more than others and whether this is significant be more meaningful (see comment on volcano plot).

83 ff: I would prefer to see also the fatty acid composition of the final products

104 ff. if the methodology for preparation of microcapsules, sausages etc. is more or less the same as previously published, these paragraphs could be much shorter. Clarify whether all products from the same raw material (meat, fat).

129 ff: as the exact recipe is not reported (e.g. amount of antioxidant used) the study cannot be replicated. This is considered a major weakness and should be reflected on by the authors.

181 ff: sentence unclear

Fig 2/3: really needed? Suggestion to increase font size because too small.

328: this is not a score but a bi-plot

605 ff. reference not complete (volume, pages…)

Reviewer 2 Report

The manuscript focuses on volatile substances in meat products with the addition of fish oil microcapsules and on the influence of these microcapsules on the storability of the tested meat products.

It is written on a professional level, Introduction provides insight into the issue, the methodology and structure of the respondents are clearly described, the results are properly presented and discussed. The conclusion summarizes the findings. The title of the article is concise and the abstract adequate. I also appreciate the large number of sources listed in the bibliography.

Detailed results of the analyzes are given in extensive tables 1 and 2. I suggest whether it would not be appropriate to present them, for example, in the form of heat maps for their better clarity.

In the future, research should be complemented by sensory evaluation of meat products to determine whether fish oil does not adversely affect their flavor and acceptability.
